# Dimethylaminododecyl Methacrylate-Incorporated Dental Materials Could Be the First Line of Defense against *Helicobacter pylori*

**DOI:** 10.3390/ijms241713644

**Published:** 2023-09-04

**Authors:** Xi Chen, Tiantian Shan, Biao Ren, Lin Zhang, Hockin H. K. Xu, Nanxi Wang, Xuedong Zhou, Hong Li, Lei Cheng

**Affiliations:** 1State Key Laboratory of Oral Diseases & National Center for Stomatology & National Clinical Research Center for Oral Diseases, West China Hospital of Stomatology, Sichuan University, Chengdu 610041, China; 2Department of Operative Dentistry and Endodontics, West China Hospital of Stomatology, Sichuan University, Chengdu 610041, China; 3Department of Advanced Oral Sciences and Therapeutics, School of Dentistry, University of Maryland, Baltimore, MD 21021, USA; 4Center for Stem Cell Biology & Regenerative Medicine, University of Maryland School of Medicine, Baltimore, MD 21201, USA; 5Marlene and Stewart Greenebaum Cancer Center, University of Maryland School of Medicine, Baltimore, MD 21201, USA; 6West China Marshall Research Center for Infectious Diseases, Center of Infectious Diseases, West China Hospital, Sichuan University, Chengdu 610041, China; 7Division of Infectious Diseases, State Key Laboratory of Biotherapy and Center of Infectious Diseases, West China Hospital, Sichuan University, Chengdu 610041, China

**Keywords:** anti-bacterial agents, *Helicobacter pylori*, biofilms, dimethylaminododecyl methacrylate

## Abstract

Oral cavity is an essential reservoir for *H. pylori*. We aimed to investigate the antibacterial effects of dimethylaminododecyl methacrylate (DMADDM) against *H. pylori*. Modified giomers were prepared by introducing 0%, 1.25% and 2.5% DMADDM monomers. Broth microdilution assay, spot assay, Alamer Blue assay, PMA–qPCR, crystal violet staining, scanning electron microscopy observation and live/dead bacterial staining were performed to evaluate the antibacterial and antibiofilm effects of DMADDM and modified giomers in vitro. Urease assay, qPCR, hematoxylin–eosin staining and ELISA were performed to evaluate the inflammation levels and colonization of *H. pylori* in vivo. In vitro experiments indicated that the minimum inhibitory concentration and minimum bactericidal concentration of DMADDM were 6.25 μg/mL and 25 μg/mL, respectively. It inhibited *H. pylori* in a dose- and time-dependent manner, and significantly reduced the expression of *cagA*, *vacA*, *flaA* and *ureB*. DMADDM-modified giomers inhibited the formation of *H. pylori* biofilm and reduced live cells within it. In vivo experiments confirmed that the pretreatment with DMADDM-modified dental resin effectively reduced the gastric colonization of oral–derived *H. pylori*, suppressed systemic and local gastric inflammation. DMADDM monomers and DMADDM-modified giomers possessed excellent antibacterial and antibiofilm effects on *H. pylori*. Pretreatment with DMADDM-modified giomers significantly inhibited the gastric infection by *H. pylori*.

## 1. Introduction

*Helicobacter pylori* is one of the most common infectious bacteria in humans [1]. It has been classified as a Class I carcinogen by the World Health Organization’s International Agency for Research on Cancer (IARC) since 1994. The infection of *H. pylori* is considered the most important risk factor for various gastric diseases such as chronic gastritis, gastric ulcer, gastric adenocarcinoma and even mucosa–associated lymphoid tissue lymphoma [2]. About 67% to 80% of duodenal ulcers are caused by *H. pylori*, and about 39% of gastric cancers are closely related to *H. pylori* infection [3]. More than half of the global population has been infected with *H. pylori* [1]. How to prevent and treat *H. pylori* infection has become a major problem all over the world.

The oral–oral route is one of the main pathways by which *H. pylori* is transmitted from person to person [4]. According to epidemiological studies, the prevalence of *H. pylori* infection was elevated within families and cohabitate populations [5]. The presence of infected parents and siblings, crowded households and bad oral habits in childhood [6] such as lack of regular teeth brushing [6,7] have all been found to be significantly and strongly associated with *H. pylori* infection. In addition, both genetic sequence and specific antigens of *H. pylori* were identified in the oral cavity [8]. Oral diseases including periodontal disease [9,10], oral leukoplakia [11] and lichen planus [12] have been reported to be related to the presence of *H. pylori*. All the evidence supports that oral cavity plays an essential role in the transmission and infection of *H. pylori* by serving as a reservoir for it [13]. The eradication rate of gastric *H. pylori* has been reported to be increased by 21% after the elimination of oral *H. pylori* [14]. The eradication of oral *H. pylori* is a beneficial supplement to the traditional prevention and treatment strategies for *H. pylori*, and oral cavity is an important but always overlooked antibacterial setting for the prevention and treatment of *H. pylori*.

The most used therapy for *H. pylori* eradication is triple or quadruple therapy involving proton pump inhibitors, antibiotics and bismuth agents. However, the eradication rate is declining year by year. Drug resistance and bacterial recurrence poses major challenges to complete pathogen eradication [15,16,17]. It has been reported that the global resistance rates of *H. pylori* to clarithromycin, metronidazole and levofloxacin have reached approximately 15% [15]. The annual recurrence, reinfection and recrudescence rate of *H. pylori* were 4.3%, 3.1% and 2.2%, respectively, in three subsequent years after the eradication treatment [18]. Oral reservoirs are attributed as one of the main causes of short–term bacterial recurrence after the eradication of gastric *H. pylori* [19,20]. However, it is difficult for traditional antibiotics and oral topical drugs to maintain long–acting antibacterial effects in the oral cavity [21,22], resulting in the inability of standard therapies to kill oral *H. pylori* effectively [23].

Dimethylaminododecyl methacrylate (DMADDM) is a potent cationic antibacterial agent with good biocompatibility [24,25]. It can penetrate cells like a needle piercing a balloon and has been shown to possess good antibacterial and antifungal effects [26]. Especially in the dental field, DMADDM can increase the proportion of antagonistic bacteria while inhibiting core microorganisms [27,28], thereby regulating the composition of dental plaque biofilm and maintaining the balance of oral microecology [29,30]. Studies on the mechanism of DMADDM drug resistance screened a variety of oral bacteria and found that it had a low risk of inducing drug resistance to common oral bacteria [31,32].

Herein, we attempted to explore the anti-*H. pylori* potential of DMADDM and build up the first line of defense against *H. pylori* infection in the oral cavity by introducing DMADDM into giomers. DMADDM can co-polymerize with dental resins, avoiding a burst release and rapid loss of active antibacterial component [33], and ensuring a potential durable contact killing effect of dental filling materials against *H. pylori*. The hypotheses tested were as follows: (1) DMADDM monomers could inhibit the growth, viability and expression of virulence factors of *H. pylori*; (2) DMADDM monomers and their modified giomers could inhibit the formation of *H. pylori* biofilms and reduce live cells within them; (3) pretreatment of *H. pylori* by DMADDM-modified giomers could reduce bacterial colonization in the stomach and inhibit systemic and local gastric inflammation in vivo.

## 2. Results

### 2.1. Inhibition of H. pylori in Planktonic Form

The minimum inhibitory concentration (MIC) and minimum bactericidal concentration (MBC) of DMADDM monomers on *H. pylori* were 6.25 μg/mL and 25 μg/mL, respectively. As shown in the spot assay results (Figure 1A), both thickness and density of *H. pylori* colonies decreased significantly when DMADDM exceeded 6.25 μg/mL; no colonies were identified when DMADDM reached 25 μg/mL. The morphology of *H. pylori* treated by different concentrations of DMADDM is depicted in Figure 1B. With the increase in DMADDM monomer concentrations, the number of *H. pylori* cells decreased significantly, accompanied by a gradual transition of cell morphology from being arc shaped in the normal state to being in the short rod or coccoid form.

DMADDM monomers significantly inhibited the growth of *H. pylori* in a concentration-dependent and time-dependent manner as shown in Figure 1C. A significant increase in the slope of OD and log_10_ CFU/mL curve was identified at both MIC and MBC of DMADDM, consistent with the results of the spot assay. Compared with the control group, DMADDM with a 4× MIC concentration achieved 87.92 ± 1.41% growth inhibition rate after 12 h incubation and 96.62 ± 0.09% after 24 h incubation (Figure 1D).

Results of the metabolism detected by an Alamar Blue kit are shown in Figure 1E. Overall, the efficiency of DMADDM in inhibiting the total metabolic activity of planktonic *H. pylori* was enhanced with increasing monomer concentration. However, after correcting the data with the number of surviving bacteria based on the results of propidium monoazide (PMA)–qPCR, the average cell metabolic activity did not decrease but increased after DMADDM treatment. When the concentration of DMADDM reached 4× MIC or higher, the average metabolism of individual surviving *H. pylori* cells was significantly higher than that of the untreated group (*p* < 0.0001).

Virulence factors of *H. pylori* play a crucial role in infection and pathogenesis. Changes in the expression of *H. pylori* virulence genes after 30 min of DMADDM treatment are shown in Figure 1F. Compared with the control group, DMADDM significantly inhibited all tested *H. pylori* virulence expressions, including *vacA*, *cagA*, *flaA* and *ureB*, even at 1/2× MIC (*p* < 0.05).

### 2.2. Inhibition of H. pylori Biofilms

From 1/4× to 8× MIC, DMADDM monomers inhibited *H. pylori* biofilm formation and eradicated biofilms in a dose-dependent manner (Figure 2A). The results of crystal violet staining and PMA–qPCR on the amount of biofilm bacteria indicated that DMADDM-modified giomers inhibited the formation of *H. pylori* biofilm (Figure 2B). With the increase in DMADDM concentration, the proportion of live bacteria within the biofilm gradually decreased (Figure 2C,D). Giomers containing 2.5% DMADDM posed stronger biofilm inhibition than those containing 1.25% DMADDM in the crystal violet staining (*p* = 0.023) and PMA–qPCR experiment (*p* = 0.001) and obtained a more significant reduction in the live bacteria proportion of biofilm in the live/dead staining (*p* = 0.024). SEM results indicated that incorporating DMADDM into giomers could reduce the density of biofilms formed in the material surface (Figure 2D).

### 2.3. Antibacterial Effects of Pretreatment In Vivo

Treatment with DMADDM-modified dental resin for 2 h significantly inhibited the gastric infection by *H. pylori*. The H&E staining revealed that gastric mucosa in the DMADDM-treated group had almost no damage, while the control-infected group showed severe inflammatory cell infiltration (Figure 3A). The qPCR (Figure 3B) and urease assay (Figure 3C) results were consistent with these, demonstrating a significant reduction in *H. pylori* gastric colonization after pretreatment of the DMADDM-modified giomers. The results of ELISA (Figure 3D) showed that the giomer treatment significantly suppressed both systemic and local gastric inflammation, even reaching the statues of healthy mice.

## 3. Discussion

The prevention and treatment of *H. pylori* has become more and more difficult due to the increasing drug resistance in recent years. The oral–oral route is one of the main routes of *H. pylori* transmission, and oral cavity is important for infection of *H. pylori* by serving as a reservoir for it. However, researchers developing new drugs have mainly focused on gastric *H. pylori*, ignoring environmental and oral *H. pylori*. This study attempted to build up the first line of defense against *H. pylori* infection by introducing DMADDM into dental filling materials. According to previous studies, the DMADDM-modified giomers maintained excellent mechanical properties, surface characterizations and biocompatibility with oral tissues [34]. Our results demonstrated that DMADDM monomers significantly inhibited the growth and viability of *H. pylori*, reduced virulence gene expression and inhibited biofilm formation. Introducing DMADDM into giomers endowed dental filling materials with good anti-*H. pylori* ability, which reduced bacterial colonization in the stomach and suppressed systemic and local gastric inflammation in animal models, highlighting its potential for preventing and treating *H. pylori* infection in the future.

After treatment with a series of concentrations of DMADDM, the surviving *H. pylori* transformed from being arc shaped in the normal state to being in coccoid form according to the SEM observation. The coccoid shape of *H. pylori* was first discovered in 1991 [35]. It was thought in the past to be a manifestation of degenerative events based on the simultaneous loss of culturability of *H. pylori* cells during morphogenesis. However, nowadays it is thought that this may be an active protective mechanism or adaptation process of *H. pylori* in response to environmental and antibiotic pressures [36]. It was speculated that in the face of an unfavorable microbial living environment, such as being treated with antisecretory antibacterial drugs and antibacterial drugs [37], *H. pylori* would transform into a specific spherical shape, corresponding to a state where the bacteria are viable but nonculturable. This endows the bacteria with a high degree of tolerance to harsh conditions and may be one of strategies for *H. pylori* to resist drug effects [38], which deserves more studies.

The killing curves revealed that DMADDM monomers inhibited the growth of *H. pylori* in a concentration- and time-dependent manner. According to SEM observation, *H. pylori* transformed into coccoid forms under high concentrations of DMADDM; this leads to the assumption that some of these bacteria entered a viable but nonculturable state [38,39]. In this case, conventional cultural enumeration methods cannot quantify viable bacteria accurately, and qPCR enumeration will be interfered with by the large amount of DNA from dead bacteria. Therefore, an assay combining the qPCR technique with PMA photosensitive dye that possessed high affinity for nucleic acids was employed in this study [40]. By penetrating dead cells and binding to their DNA, PMA inhibited the PCR amplification of dead cells, allowing the selective quantification of viable bacteria [41]. Compared to traditional methods, PMA–qPCR may be a more accurate choice to enumerate viable bacteria and may also serve as a potential tool to explore the relationship between bacterial survival state and morphological transition.

From the metabolism results, DMADDM significantly inhibited the total metabolism of *H. pylori* in the culture media. After correcting the results with the number of viable cells, it was found that the metabolism of the survived *H. pylori* was significantly enhanced, which may be a stress response or survival mechanism for drug-resistant bacteria to survive high–concentration drug interventions [42,43]. Deepening research in this field will help to develop new anti-*H. pylori* drugs, reducing drug resistance.

According to the virulence gene expression assay, treatment of DMADDM for 30 min effectively reduced the expression levels of *vacA*, *cagA*, *flaA* and *ureB* genes. While VacA and CagA are two main virulence factors of *H. pylori* in causing host tissue damage [44,45], *flaA* and *ureB* genes are responsible for promoting the rapid passage of *H. pylori* through the gastric mucous layer and converting urea into ammonia to facilitate survival and persistence of *H. pylori* in the harsh gastric environment [46,47]. By inhibiting key virulence factors, DMADDM may impair the adaptation of *H. pylori* to the gastric environment including gastric acid and the mucus layer. However, whether this rapid inhibition directly leads to a reduced ability of bacteria to colonize the stomach warrants further studies.

To investigate the antibiofilm effects of DMADDM-modified giomers, SEM morphological observation, crystal violet staining, live/dead staining and PMA–qPCR viable bacterial quantification were performed in the present study. Based on our results, the formation of *H. pylori* biofilm on the material surface was significantly inhibited, accompanied by a decreased density and an increase in the proportion of dead bacteria within the biofilm. This indicated that the immobilization of DMADDM by dental resin materials maintained its anti-*H. pylori* capability. From our results, incorporating 2.5% DMADDM into giomers posed a better antibacterial effect than 1.25% DMADDM, and it has been reported that giomers containing 2.5% DMADDM have a better mechanical performance than 5% DMADDM and could inhibit cariogenic biofilm to provide caries protection from a dental perspective [34]. Therefore, 2.5% is the proposed concentration when incorporating DMADDM into dental materials.

In vivo experiments further confirmed that the pretreatment with DMADDM-modified dental resin effectively reduced the gastric colonization of oral-derived *H. pylori*, and suppressed the systemic and local gastric level of IL–1β, IL–6 and TNF–α. The abundance of *H. pylori* in the oral cavity is lower than the concentration in vivo experiments and in terms of processing time, *H. pylori* exists stably in dental plaque for a long time and DMADDM cross-linked resin can also play a long-term stable contact antibacterial effect. We simplified this long-term process to 2 h and have achieved a good antibacterial effect. These results suggested that the use of anti-*H. pylori* dental materials could block the oral transmission of *H. pylori*, and thus played an important role in the prevention of gastritis.

Since *H. pylori* is considered to infect individuals in their childhood, it is not clear if DMADDM could protect the individuals from the initial infection. However, the findings of this study suggest that DMADDM could be protective against the re-infection after *H. pylori* eradication therapy. Within the limitations of the present study, it may be concluded that DMADDM monomers possess strong anti-*H. pylori* activity with their low MIC and MBC, and the DMADDM-modified dental filling materials effectively inhibit *H. pylori* biofilm formation, reduce gastric colonization of the bacteria and suppress systemic and local gastric inflammation in vivo. Therefore, the present results imply that dental filling materials containing anti-*H. pylori* agents, such as DMADDM, have great potential in serving as the first line of defense in oral cavity to prevent and treat *H. pylori* infections.

## 4. Materials and Methods

### 4.1. Synthesis of DMADDM and DMADDM–Modified Giomers

DMADDM was synthesized via a modified Menschutkin reaction method [48]. Briefly, 1–(dimethylamino)docecane (DMAD) (Tokyo Chemical Industry, Tokyo, Japan) and 2–bromoethyl methacrylate (BEMA) (Monomer–Polymer and Dajac Labs, Trevose, PA, USA) were first mixed at a 1:1 concentration of substance with a magnetic stirring. The mixture was stirred and reacted at 70 °C for 24 h. After removing the ethanol solvent by evaporation, DMADDM was obtained as a colorless and viscous liquid. To impart antimicrobial capacity to dental filling materials, DMAMM was then introduced into the commercial giomers (Beautifil II F03 A2, SHOFU Inc., Kyoto, Japan) at mass fraction of 0%, 1.25% and 2.5% DMADDM (m/m). The commercial giomers without DMADDM served as the control.

### 4.2. H. pylori Strains and Biofilm Cultivation

The *H. pylori* G27 strain was kindly provided by Professor Hong Li from West China Hospital of Sichuan University and grown on the Columbia blood agar (CBA) (Oxoid, Basingstoke, UK) plates at 37 °C under microaerophilic conditions (5% oxygen, 10% carbon dioxide and 85% nitrogen) [49]. The bacterial inoculation was adjusted approximately to 1.0 × 10^6^ CFU/mL and incubated in brain heart infusion (BHI) broth (Becton, Dickinson and Company, Franklin Lakes, NJ, USA), supplemented with fetal bovine serum (FBS) (Gibco, Grand Island, NY, USA) for 48–72 h to form biofilms [50].

### 4.3. Antimicrobial Activity Assay

The MIC of the DMADDM monomers against *H. pylori* was determined by a standard broth microdilution assay following the guidelines of the Clinical and Laboratory Standards Institute [51]. The suspension of *H. pylori* was adjusted to 1.0 × 10^5^ CFU/mL. DMADDM was added to the medium at a concentration from 1.5125 μg/mL to 50 μg/mL. The group without DMADDM was determined as the blank control. All groups were adjusted to 100 μL with bacterial suspension and incubated in the 96–well plate at 37 °C. The MIC endpoint was defined as the lowest concentration at which no visible bacterial growth appeared after 24 h.

MBC was defined as the lowest concentration of antibiotic in which more than 99.9% of the bacteria were killed compared with a blank control [52]. Therefore, bacterial suspensions above MIC were sub-cultured on the surface of CBA plates. The MBC was defined as the lowest concentration at which no colonies formed after 48 h of incubation. All the assays were performed in triplicate.

### 4.4. Morphology Changing Assay

Scanning electron microscopy (SEM) was used to observe the morphological changes of *H. pylori* treated with different concentrations of DMADDM (1/8×, 1/4×, 1/2×, 1×, 2×, 4× and 8× MIC). The bacterial suspension was rinsed with PBS (Gibco, Grand Island, NY, USA), fixed overnight by 2.5% glutaraldehyde, immobilized on a sterile glass disc, kept in a desiccator for 24 h and dehydrated with an ascending serious of ethanol agents. All specimens were examined with SEM (Quanta 200, FEI Company, Hillsboro, OR, USA) after gold spraying.

### 4.5. Bacterial Killing Assay

The bacterial inoculation was adjusted to approximately 1.0 × 10^5^ CFU/mL and co-cultured with different concentrations (1×, 2×, 4× and 8× MIC) of DMADDM monomers. The optical densities (OD) of bacterial inoculation in all groups at different time points (0, 1, 2, 3, 6, 12 and 24 h) were measured at 600 nm with SpectraMax iD5 (Molecular Devices, Sunnyvale, CA, USA) and normalized with the media-only substrate control.

*H. pylori* treated with different concentrations of DMADDM and for different times was enriched by centrifugation and resuspended in PBS. PMA (US EVERBRIGHT, Jiangsu, China) was added to the bacterial solution in the dark and incubated for 5 min at room temperature on a shaker. Afterwards, the specimen was exposed to bright light for 5 to 10 min to allow for the full cross-linking of PMA to the extracellular DNA. Next, DNA within live bacteria was extracted according to the instructions of the bacterial genomic DNA extraction kit (TIANGEN, Beijing, China) [53]. Using the extracted DNA as a template, qPCR detection was performed with the TB Green™ Premix Ex Taq™ II kit (Takara, Kyoto, Japan). Primers utilized in the study are shown in Table 1. All the assays were performed in triplicate.

### 4.6. Viability Test

A suspension of approximately 1.0 × 10^5^ CFU/mL of *H. pylori* supplemented with various concentrations of DMADDM (1/2×, 1×, 2×, 4× and 8× MIC). After culturing for 24 h, bacterial suspension from each group was mixed with the Alamar Blue (Alamar Blue cell proliferation/cytotoxicity detection kit, BestBio, Shanghai, China) and incubated for 6 h. The fluorescence intensity was identified by SpectraMax iD5. The excitation and emission wavelength were 560 nm and 590 nm, respectively. The experiment was conducted in triplicate.

### 4.7. Virulence Gene Expression Assay

A suspension of approximately 1.0 × 10^6^ CFU/mL of *H. pylori* supplemented with various concentrations of DMADDM (1/2×, 1×, 2×, 4× and 8× MIC) was incubated for 30 min. Total RNA from cells was isolated using TRIzol Reagent (Invitrogen, Carlsbad, CA, USA). The PrimeScript™ RT reagent kit with the gDNA Eraser kit (Takara, Kyoto, Japan) was used for RNA reverse transcription. Primers targeted to *vacA*, *cagA*, *flaA* and *ureB* genes and *16S rRNA* (Table 1), the housekeeping gene of *H. pylori*, were examined. The qPCR detection was performed with TB Green^TM^ I kit (Takara, Kyoto, Japan).

### 4.8. Biofilm Inhibiting Assay

To evaluate the inhibitory effect of DMADDM on *H. pylori* biofilm formation, various concentrations of DMADDM were co-cultured with the bacteria for 48–72 h before biofilm formation. To evaluate the biofilm removal capability of DMADDM, different concentrations of DMADDM were applied to the biofilm for 24 h. Crystal violet staining was performed, and the OD was measured at 595 nm.

### 4.9. Antibiofilm Assays of DMADDM–Modified Giomers

Bacterial suspension with 1.0 × 10^6^ CFU/mL *H. pylori* was treated with giomers containing 0%, 1.25% and 2.5% DMADDM for 48–72 h to develop bacterial biofilm. SEM observation, crystal violet staining and PMA–qPCR were performed as aforementioned. Live/dead staining was performed with a premixed solution containing Syto9, propidium iodide (PI; Kit L 7012, Invitrogen, Carlsbad, CA, USA) and 85% NaCl at a ratio of 1:1:98 for 15 min. All specimens were observed with confocal laser scanning microscopy (CLSM; LSM 710, Zeiss, Oberkochen, BW, Germany). The excitation wavelengths were 488 nm for SYTO 9 and 543 nm for PI.

### 4.10. In Vivo Inhibition Efficacy of DMADDM–Modified Giomers against H. pylori Infection

All animal experiments were approved by the Ethics Committee of West China Hospital of Stomatology of Sichuan University (No. WCHSIRB–D–2022–129). Four–week–old C57BL/6 male mice were randomly divided into three groups (*n* = 6) and oral gavaged every other day for 4 times with 200 μL of sterile PBS (negative control group) or 10^9^ CFU/mL *H. pylori*, that were treated with or without 2.5% DMADDM-modified giomers for 2 h. One week after infection, all mice were sacrificed, and the stomach was removed from the abdominal cavity [58]. One half of the stomach section was used to assess bacterial colonization by urease assay and qPCR, and the other half was used for hematoxylin–eosin (H&E) staining and to determine the levels of IL–1β, IL–6 and TNF–α in the stomach by ELISA [59]. Serum samples were also collected and analyzed for IL–1β, IL–6 and TNF–α levels by ELISA to determine systemic inflammation in the mice [59].

### 4.11. Statistical Analysis

In the present study, the normality and equal variance assumption were analyzed by Shapiro–Wilk test and modified Levene test, respectively. One–way analysis of variance (ANOVA) with Tukey’s multiple comparisons was performed to compare the data. All statistical analyses were carried out at a significance level of 5% using the IBM SPSS Statistics 25.0.0.2 software (SPSS, Chicago, IL, USA).

## 5. Conclusions

DMADDM had an excellent inhibitory effect on *H. pylori* both in vitro and in vivo. DMADDM monomers possessed strong antibacterial activity with their low MIC and MBC, significantly inhibited the growth as well as proliferation of *H. pylori*, and had a significant inhibitory effect during and after the formation of *H. pylori* biofilm. Introducing DMADDM into giomers successfully imparted the anti-*H. pylori* biofilm ability to the dental filling material. DMADDM-modified dental materials reduced gastric colonization of the bacteria and suppressed systemic and local gastric inflammation in vivo, which may be served as the first line of defense against *H. pylori* in the oral cavity (Figure 4).

## Figures and Tables

**Figure 1 ijms-24-13644-f001:**
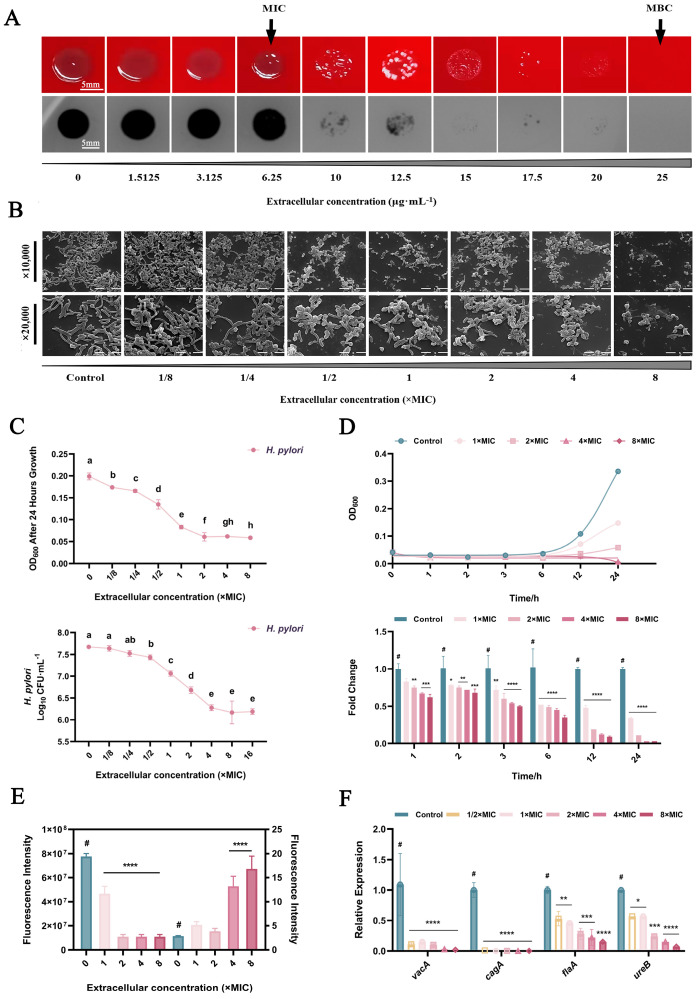
Antibacterial effects of DMADDM monomers on *H. pylori* at different concentrations. (**A**) Growth of *H. pylori* colonies treated with DMADDM monomers at different concentrations. (**B**) SEM results of the morphological changes of DMADDM–treated *H. pylori*. (**C**) Optical density values and PMA–qPCR results of *H. pylori* under different concentrations of DMADDM monomers after 24 h incubation. Values with different lowercase letters were significantly different (*p* < 0.05) (**D**) Optical density values and PMA–qPCR results of *H. pylori* treated with different concentrations and time of DMADDM monomers. (**E**) Inhibitory effects of DMADDM on the metabolic activity of *H. pylori*. (**F**) Levels of mRNA of *H. pylori* virulence genes under different concentrations of DMADDM monomers. Data are presented as means ± standard deviations. Data marked with an asterisk are significantly different from the control group. (#, the control group; * *p* < 0.05, ** *p* < 0.01, *** *p* < 0.001, **** *p* < 0.0001).

**Figure 2 ijms-24-13644-f002:**
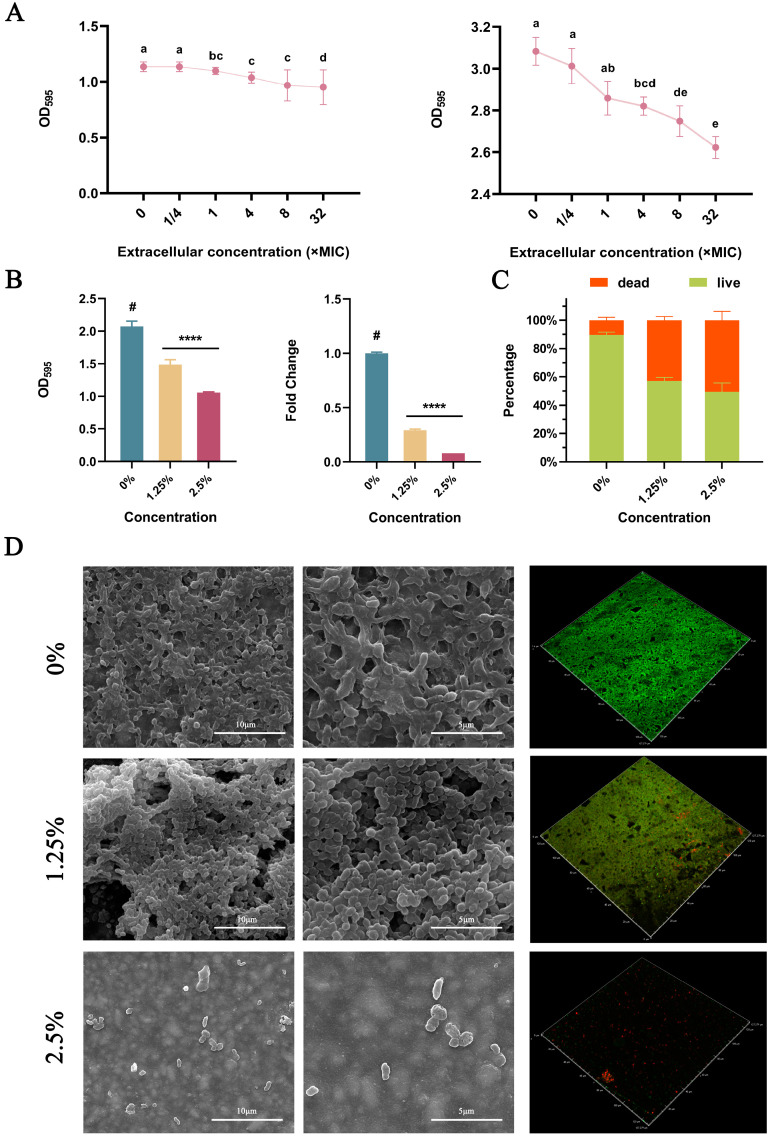
Anti–biofilm effects of DMADDM monomers and DMADDM-modified giomers on *H. pylori*. (**A**) Crystal violet staining of *H. pylori* cultured with DMADDM before (the left) and after (the right) biofilm formation. Values with different lowercase letters were significantly different (*p* < 0.05). (**B**) Crystal violet staining and PMA–qPCR results of biofilms cultured on giomers containing different concentrations of DMADDM. Data are presented as means ± standard deviations (#, the control group; **** *p* < 0.0001). (**C**) The dead/live cell ratios of *H. pylori* biofilm cultured on giomers containing different concentrations of DMADDM. (**D**) SEM and live/dead staining results of *H. pylori* biofilm cultured on DMADDM-modified giomers.

**Figure 3 ijms-24-13644-f003:**
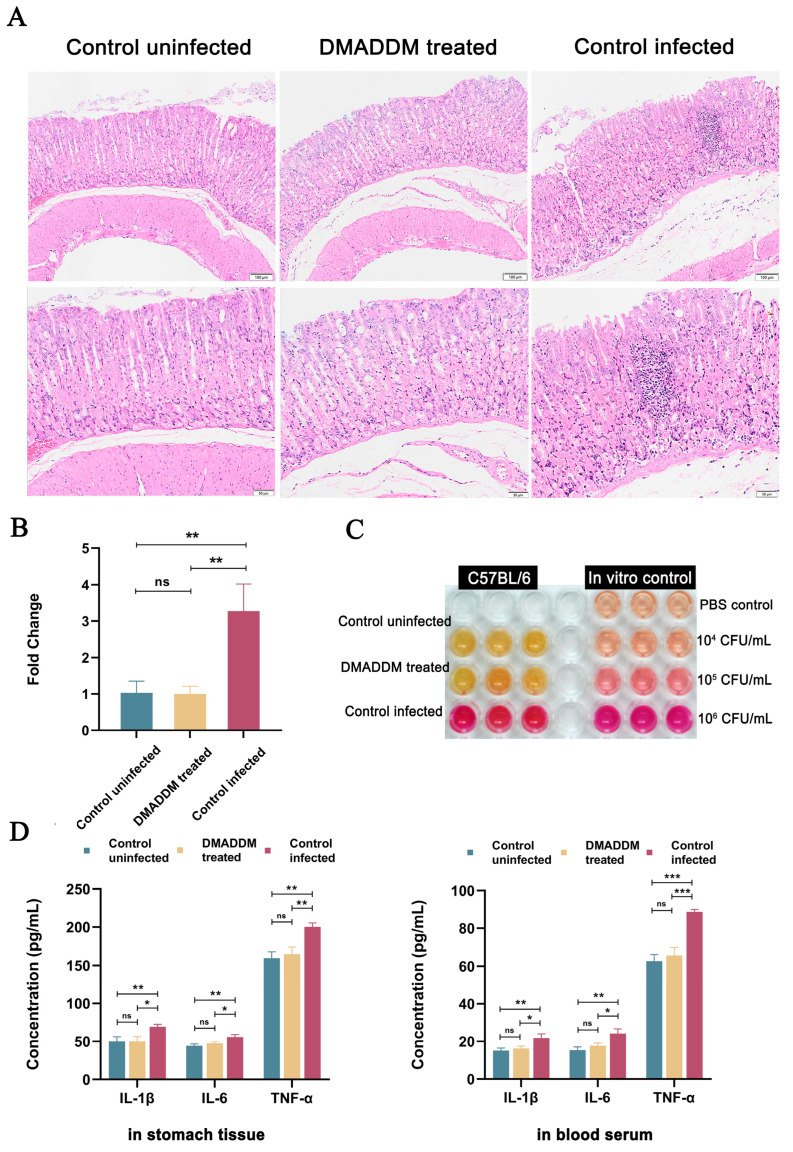
In vivo inhibition efficacy of DMADDM-modified giomers against *H. pylori* infection. (**A**) Hematoxylin–eosin staining results of gastric mucosa infected with PBS and *H. pylori* that were treated with or without DMADDM–modified giomers. (**B**) The qPCR results assessing *H. pylori* colonization in the stomach. Data are presented as means ± standard deviations. (ns, no significant difference; ** *p* < 0.01). (**C**) Urease assay results assessing *H. pylori* colonization in the stomach. (**D**) ELISA analysis for the IL–1β, IL–6 and TNF–α levels in the stomach and serum samples. Data are presented as means ± standard deviations. (ns, no significant difference; * *p* < 0.05, ** *p* < 0.01, *** *p* < 0.001).

**Figure 4 ijms-24-13644-f004:**
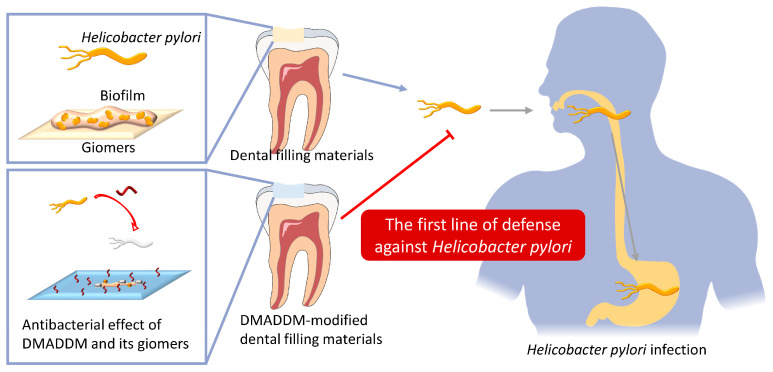
Both DMADDM monomers and DMADDM-modified giomers possessed excellent anti-*H. pylori* capability. The oral cavity can theoretically become an important antibacterial setting for the prevention and treatment of *H. pylori*, and the DMADDM-modified dental filling materials are expected to be the first line of defense against *H. pylori* in the oral environment.

**Table 1 ijms-24-13644-t001:** Specifications of primers used for quantitative PCR.

Gene	5′–3′ Sequence	Product Size (bp)
*vacA* [54]	Fwd: CCTACTGAGAATGGTGGCAATA Rvs: GTTCTTCACGAGAGCGTAGTT	87
*cagA* [54]	Fwd: GACCGACTCGATCAAATAGCA Rvs: TTAGCTGAAAGCCCTACCTTAC	113
*flaA* [55]	Fwd: CAGTATAGATGGTCGTGGGATTG Rvs: GAGAGAAAGCCTTCCGTAGTTAG	127
*ureB* [56]	Fwd: CAAAATCGCTGGCATTGGT Rvs: CTTCACCGGCTAAGGCTTCA	100
*16S rRNA* ^1^, [57]	Fwd: CTCATTGCGAAGGCGACCT Rvs: TCTAATCCTGTTTGCTCCCCA	76

^1^ Housekeeping gene; Fwd, Forward; Rvs, Reverse.

## Data Availability

Not applicable.

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
