# Peer review of "Dimethylaminododecyl Methacrylate-Incorporated Dental Materials Could Be the First Line of Defense against *Helicobacter pylori"

_ijms, 2023, doi:10.3390/ijms241713644_

Round 1
Reviewer 1 Report
The manuscript titled ‘Dimethylaminododecyl methacrylate-incorporated dental materials - The first line of defense against Helicobacter pylori in oral cavity’ explored the antimicrobial and antibiofilm properties of DMADDM and DMADDM-modified giomers against H. pylori. The antimicrobial experiments conducted were comprehensive and robust. However, some aspects remain unclear from the results, particularly regarding the efficacy of DMADDM-modified giomers as a restorative dental material in inhibiting H. pylori. Specific concerns raised are as follows:
1. DMADDM monomers and DMADDM-modified giomers showed antimicrobial effects in a dose-dependent manner. What is the optimal concentration of DMADDM the authors would propose when incorporating DMADDM into dental materials?
2. Although the in vivo experiments demonstrated that mice orally gavaged with DMADDM-modified giomer-treated H. pylori showed less systemic and local gastric inflammation, how did these findings relate to the practical scenario? Specifically, when DMADDM-modified giomers are used for tooth restoration, the amount of DMADDM in the oral cavity and the duration of interaction between DMADDM and H. pylori may differ significantly from the conditions in the present in vivo experiment. The authors should address how these results can be applied practically.
3. What is the concentration of DMADDM used in the in vivo study?
4. Instead of using generic labels like 'negative control' and 'positive control' in the in vivo study images, I suggest the authors use specific treatment labels for clarity.
5. It is recommended to provide a clear definition of MIC and MBC at their first appearance in the manuscript to ensure readers' understanding.
6. English editing is suggested. Typos and grammar errors were found in the manuscript, e.g., Line 175, ‘oral cavitary’; Line 138, ‘…in a dose-dependent (Figure 2A)…’
English can be improved.
Author Response
Dear Editors and Reviewers,
Thank you very much for reviewing the manuscript. We must express our sincere gratitude for your meticulous feedback. All the comments were very useful, and we have carefully revised our manuscript. Please find our point-to-point responses to your questions and comments below.
Comments: (1) DMADDM monomers and DMADDM-modified giomers showed antimicrobial effects in a dose-dependent manner. What is the optimal concentration of DMADDM the authors would propose when incorporating DMADDM into dental materials?
Response: Thank you very much. According to our results, giomers containing 2.5% DMADDM posed stronger biofilm inhibition than that containing 1.25% DMADDM in the crystal violet staining (p=0.023) and PMA-qPCR experiment (p=0.001), obtained more significant reduction of live bacteria proportion in biofilm in the live/dead staining (p=0.024). Besides, the research by Chen et al. (Chen, et al. 2022) showed that giomers containing 2.5% DMADDM has a better mechanical performance than 5% DMADDM and could inhibit cariogenic biofilm to provide caries protection from a dental perspective. Therefore, 2.5% is the proposed concentration when incorporating DMADDM into dental materials. Modifications has been made in the revised manuscript.
References
Chen Y., Yang B., Cheng L., Xu H.H.K., Li H., Huang Y., Zhang Q., Zhou X., Liang J. , Zou J., Novel Giomers Incorporated with Antibacterial Quaternary Ammonium Monomers to Inhibit Secondary Caries. Pathogens. 2022;11(5). doi:10.3390/pathogens11050578
Comments: (2) Although the in vivo experiments demonstrated that mice orally gavaged with DMADDM-modified giomer-treated H. pylori showed less systemic and local gastric inflammation, how did these findings relate to the practical scenario? Specifically, when DMADDM-modified giomers are used for tooth restoration, the amount of DMADDM in the oral cavity and the duration of interaction between DMADDM and H. pylori may differ significantly from the conditions in the present in vivo experiment. The authors should address how these results can be applied practically.
Response: Thank you for pointing this out. Oral cavity plays an essential role in the transmission and infection of H. pylori by serving as a reservoir for it. There are more than 400 species of microorganisms in the oral cavity, the number of which is as high as tens of billions or even hundreds of billions, while the abundance of H. pylori is actually very low. In the in vivo experiments, we used a bacterial concentration of 109 CFU/mL, which is much higher than the actual concentration in the oral cavity, but can be used to test the strategy of employing DMADDM-modified dental materials to prevent H. pylori infection in the oral cavity. According to the in vivo experiments, 2.5% DMADDM modified resin material reduced the short-term colonization and infection level of H. pylori at a concentration of 109 CFU/mL to the same level (p>0.05) as that of uninfected mice, which is fully capable of coping with the practical scenario.
In terms of processing time, studies have shown that H. pylori exists stably in dental plaque for a long time (Anand, et al. 2014), and DMADDM cross-linked resin can also play a long-term stable contact antibacterial effect (Yu, et al. 2020). We simplified this long-term process to 2 hours and have achieved good antibacterial effect. Wang et al. (Wang, et al. 2017) have reported that DMADDM is not easy to induce bacterial resistance, which is also the content of our follow-up research. Modifications has been made in the revised manuscript.
References
Anand P.S., Kamath K.P. , Anil S., Role of dental plaque, saliva and periodontal disease in Helicobacter pylori infection. World J Gastroenterol. 2014;20(19):5639-5653. doi:10.3748/wjg.v20.i19.5639
Yu J., Huang X., Zhou X., Han Q., Zhou W., Liang J., Xu H.H.K., Ren B., Peng X., Weir M.D., et al, Anti-caries effect of resin infiltrant modified by quaternary ammonium monomers. J Dent. 2020;97103355. doi:10.1016/j.jdent.2020.103355
Wang S., Wang H., Ren B., Li H., Weir M.D., Zhou X., Oates T.W., Cheng L. , Xu H.H.K., Do quaternary ammonium monomers induce drug resistance in cariogenic, endodontic and periodontal bacterial species? Dent Mater. 2017;33(10):1127-1138. doi:10.1016/j.dental.2017.07.001
Comments: (3) What is the concentration of DMADDM used in the in vivo study?
Response: As we replied in comment (1), giomers containing 2.5% DMADDM obtains significantly better antibacterial effect than 1.25% DMADDM, and it also poses a better mechanical performance than 5% DMADDM. The concentration of DMADDM used in the in vivo study is 2.5%, and the manuscript has been revised.
Comments: (4) Instead of using generic labels like 'negative control' and 'positive control' in the in vivo study images, I suggest the authors use specific treatment labels for clarity.
Response: Thank you very much. It has been revised in the manuscript and figures.
Comments: (5) It is recommended to provide a clear definition of MIC and MBC at their first appearance in the manuscript to ensure readers' understanding.
Response: Thank you very much. It has been revised in the manuscript.
Comments: (6) English editing is suggested. Typos and grammar errors were found in the manuscript, e.g., Line 175, ‘oral cavitary’; Line 138, ‘…in a dose-dependent (Figure 2A)…’
Response: Thank you very much. The typos and grammar errors have been revised. In addition, we have carefully checked and revised the typos and grammar of the entire article.
Reviewer 2 Report
It was a pleasure to review this interesting and relevant paper. It is well written and has clinical importance. I can only offer two minor amendments due to spelling or grammar errors:
Line 175 - cavity
Line 176 - researchers
I have approached this review from a clinical perspective and application into the clinical setting. While the paper outlines a potential application it is accepted that this will take additional research and product development. It is also accepted that the clinical benefit would apply only to adults patients who need dental restorations and therefore would have no direct clinical benefit for those with high standard of oral hygiene who, as explained in the papers introduction, would be at lower risk of H.Pylori gastritis). The merit of this paper is that it adds to existing literature on this subject. It is accepted that the oral cavity is a reservoir for H.pylori and therefore contributes to gastritis and gastric ulceration. What this paper adds is the novel addition of DMADDM into conventional dental restorations. Further biomechanical research would be needed prior to clinical trials to ensure this additions doesn't impact on the tensile strength, wear pattern or handling, but it is a fascinating concept that has shown a reduction in gastric changes in vivo. The methodology appears sound, the data well presented, and suggests a potential clinical application. On reflection, the discussion could be strengthened by considering future clinical application and next steps as this hasn't been covered in detail however that wasn't the stated aim of the paper. Overall I feel this is a good paper, well written and covers the stated aims.

No concerns. Well written. Spelling errors highlighted in attached paper.
Author Response
Dear Editors and Reviewers,
Thank you very much for reviewing the manuscript. We must express our sincere gratitude for your meticulous feedback. All the comments were very useful, and we have carefully revised our manuscript. The typos and grammar errors have been revised. In addition, we have carefully checked and revised the typos and grammar of the entire article.
Round 2
Reviewer 1 Report
The authors have addressed all my concerns. However, proofreading of the manuscript is necessary. I noticed several typos and grammatical errors, for example:
Line 145, '...ontained more significant...'. It should be '...and obtained more significant...'
Line 241, ‘...giomers containing 2.5% DMADDM has a…’. It should be ‘...have a…’
Line 248, ‘… much lower than the concentration in vivo experiments…’. It should be ‘…lower than the concentration in in vivo experiment…’
Proofreading is necessary.
Author Response
Dear Editors and Reviewers,
Thank you very much for reviewing the manuscript. We must express our sincere gratitude for your meticulous feedback. We have carefully revised our manuscript. The typos and grammar errors have been revised.